# DnaJC7 in Amyotrophic Lateral Sclerosis

**DOI:** 10.3390/ijms23084076

**Published:** 2022-04-07

**Authors:** Allison A. Dilliott, Catherine M. Andary, Meaghan Stoltz, Andrey A. Petropavlovskiy, Sali M. K. Farhan, Martin L. Duennwald

**Affiliations:** 1Department of Neurology and Neurosurgery, McGill University, Montréal, QC H4A 3J1, Canada; allison.dilliott@mcgill.ca (A.A.D.); sali.farhan@mcgill.ca (S.M.K.F.); 2Department of Anatomy and Cell Biology, Schulich School of Medicine and Dentistry, University of Western Ontario, London, ON N6A5C1, Canada; candary3@uwo.ca (C.M.A.); mstoltz@uwo.ca (M.S.); apetrop5@uwo.ca (A.A.P.); 3Department of Human Genetics, McGill University, Montréal, QC H4A 3J1, Canada

**Keywords:** *DNAJC7*, amyotrophic lateral sclerosis, molecular chaperones, J proteins, protein misfolding, Hsp70, Hsp90

## Abstract

Protein misfolding is a common basis of many neurodegenerative diseases including amyotrophic lateral sclerosis (ALS). Misfolded proteins, such as TDP-43, FUS, Matrin3, and SOD1, mislocalize and form the hallmark cytoplasmic and nuclear inclusions in neurons of ALS patients. Cellular protein quality control prevents protein misfolding under normal conditions and, particularly, when cells experience protein folding stress due to the fact of increased levels of reactive oxygen species, genetic mutations, or aging. Molecular chaperones can prevent protein misfolding, refold misfolded proteins, or triage misfolded proteins for degradation by the ubiquitin–proteasome system or autophagy. DnaJC7 is an evolutionarily conserved molecular chaperone that contains both a J-domain for the interaction with Hsp70s and tetratricopeptide domains for interaction with Hsp90, thus joining these two major chaperones’ machines. Genetic analyses reveal that pathogenic variants in the gene encoding DnaJC7 cause familial and sporadic ALS. Yet, the underlying ALS-associated molecular pathophysiology and many basic features of DnaJC7 function remain largely unexplored. Here, we review aspects of DnaJC7 expression, interaction, and function to propose a loss-of-function mechanism by which pathogenic variants in *DNAJC7* contribute to defects in DnaJC7-mediated chaperoning that might ultimately contribute to neurodegeneration in ALS.

## 1. Introduction

ALS is an incurable neurodegenerative disease associated with protein misfolding and impaired cellular protein quality control, which leads to progressive loss of motor neurons and, thus, motor function. Classic cases display a median survival period of only 2–4 years [1]. On the cellular level, ALS is characterized by misfolding, mislocalization, and inclusion formation of RNA-binding proteins such as TAR DNA-binding protein 43 (TDP-43), fused in sarcoma (FUS), Matrin3 (MATR3), and superoxide dismutase (SOD1) [2]. Most ALS cases (~90%) occur sporadically (sALS) with contributing genetic and environmental factors remaining mostly unknown. By contrast, ~10% of ALS cases are familial (fALS), presumed to be caused by pathogenic genetic variants, generally occurring in one of more than 30 genes.

We and then others found that pathogenic variants within the gene *DNAJC7* (*TPR2*; tetratricopeptide repeat 2) are associated with ALS in a study comprising nearly 4000 ALS cases and 8000 ethnically matched controls [3]. The gene encodes a protein belonging to the J protein/Hsp40 family, a subclassification of molecular chaperones imperative for protein quality control by maintaining protein function and preventing toxicity often associated with protein misfolding. Therefore, the association between *DNAJC7* genetic variants and ALS has motivated our postulation that in addition to the impaired RNA processing, vesicle trafficking, and transposon suppression known to contribute to cellular toxicity and, ultimately, neurodegeneration in ALS, protein misfolding and impaired protein quality control are early and fundamental features of ALS pathogenesis.

In this review article, we focus on the function and disease association of *DNAJC7*. We review the evolutionary lineage, expression pattern, and transcriptional regulation of *DNAJC7* as well as the interaction of DnaJC7 with Hsp70 and Hsp90 chaperones and its proposed functions in cellular protein quality control. We discuss the most recent evidence for the involvement of DnaJC7 in ALS and propose plausible mechanisms by which pathogenic variants in *DNAJC7* may contribute to ALS, focusing on its chaperone function.

## 2. J Proteins

As previously described, J proteins, also known as Hsp40s, constitute a special class of molecular chaperones involved in protein quality control. They are the most numerous and the most diverse molecular chaperone class in eukaryotic cells with 22 different J proteins expressed in yeast (*S. cerevisiae*), 49 in humans, and a staggering 89 in the plant *A. thaliana* [4,5]. This genetic diversity likely arose through gene duplications, which subsequently allowed J proteins to functionally diversify. J proteins were initially regarded as interchangeable isoforms that merely functioned to drive Hsp70 ATP hydrolysis; however, this view has evolved due to the fact of several discoveries demonstrating the immense functional diversity of J proteins and their role as highly specialized mediators of cellular protein quality control [6,7].

All J proteins contain a ~70 amino acid long J-domain that consists of three alpha-helices and a histidine, proline, and aspartic acid (HPD) motif between helices II and III [8]. Although the J-domain sequence varies among J proteins, the HPD motif is highly conserved and is key to the interaction with Hsp70. The general function of J proteins is mediated by their J-domain and was first deciphered in vitro using purified DnaJ from *E. coli*. Liberek et al. first documented that DnaJ increases the otherwise weak ATPase activity of DnaK (Hsp70) [9]. Ensuing studies showed that binding of DnaJ to Hsp70 is essential for Hsp70-mediated substrate folding, that the J-domain is essential for this function, and that DnaJ recruits client proteins to Hsp70 [10,11,12]. Together, these basic biochemical studies unraveled the J protein/Hsp70 ATPase cycle, a common example of a chaperone cycle [13]. In this cycle, the J protein binds to the client protein, transfers it to the Hsp70 substrate binding domain, and activates the Hsp70 ATPase. Then the helical lid of Hsp70 closes and the client protein is folded. Finally, after a conformational change induced by nucleotide exchange, the J protein dissociates from Hsp70, and the client protein is released from Hsp70.

J proteins are subdivided into three structural classes, all of which contain the compact helical J-domain [14]. Type I J proteins (DNAJAs) are unique due to the fact of their cysteine-rich zinc finger domain, which is linked to the J-domain by a glycine/phenylalanine-rich region. The subtype also contains a C-terminal domain hypothesized to be involved in client protein binding and primarily involved in mitigating proteotoxic stress by facilitating protein folding, refolding, and degradation [15,16]. In contrast, Type II J proteins (DNAJBs) lack the cysteine-rich region, rather containing a J-domain linked by glycine/phenylalanine-rich region to the C-terminal domain. DNAJBs are uniquely suited to aid in protein disaggregation [17]. Finally, Type III J proteins (DNAJCs) contain only the signature J-domain without the remaining features of DNAJAs and DNAJBs but are otherwise highly diverse displaying a wide range of cellular functions including those unrelated to their Hsp70 co-chaperone functions [18].

Given their central role in cellular protein quality control, including protein folding and clearance of misfolded proteins, the involvement of J proteins in neurodegenerative diseases resulting from protein aggregation and accumulation is unsurprising. Additionally, the overexpression of J proteins has displayed neuroprotective effects [3], including suppressed protein aggregation and accumulation of reactive oxidative species [17], decreased α-synuclein aggregation [19], improved motor neuron survival [20], and prevention of tau aggregation [21], which will be discussed further below. Table 1 outlines all J proteins that have been previously reported to be associated with a neurological phenotype. Although this review focuses on the discussion of DnaJC7 and its association with ALS, based on the abundance of associations between J proteins and various neurological conditions, it will remain important to continue exploring the involvement of other J proteins in the neuropathological pathways of diseases such as neurodegeneration.

## 3. DnaJC7

DnaJC7, otherwise referred to as Tpr2 or CCRP, is a conserved, albeit unusual, type of J protein that is highly expressed within the brain [3]. Although it does contain the expected J-domain, DnaJC7 is unique, as it contains two tetratricopeptide (TPR) domains, TPR1 and TPR2, which are suggested to allow DnaJC7 to bridge the two key molecular chaperone systems Hsp90 and Hsp70 (Figure 1) [22,23]. Specifically, DnaJC7 has been found to modulate the flow of substrates from Hsp70 to Hsp90, providing a control mechanism in proteins such as glucocorticoids and progesterone receptors [23,24]. The J protein has also demonstrated retention and co-activating capabilities with the constitutive active/androstane receptor (CAR), suggesting an important role in CAR-mediated gene activation [25].

Overall, Hsp70s and J proteins are among the most conserved chaperone systems throughout evolution, particularly among metazoans, likely due to the fact of their essential role in assisting the folding or disassembly of protein structures and protein homeostasis generally. These essential functions ensure conservation of protein-interacting domains of the molecular chaperones and their co-chaperones, which is especially obvious for DNAJ proteins [13]. *DNAJC7* is no exception to this level of conservation, with orthologs containing both the classical J-domain for its interactions with the Hsp70 system as well as its two highly conserved TPR domains [21]. DnaJC7 orthologues can be found in all higher chordates and even most vertebrates including birds, alligators, turtles, lizards, mammals, amphibians, coelacanths, bony fishes, lampreys, and the cartilaginous fishes (Figure 2) [30].

Interestingly, the arrangement of the domains of DnaJC7, specifically the TPR domains, are also shared with other proteins. One such protein is the Hsp70–Hsp90-organizing protein (Hop), a co-chaperone that also binds Hsp70 and Hsp90 to facilitate substrate transfer [31]. The similarities between the TPR domain organization between DnaJC7 and Hop has led to previous propositions that the two proteins possess shared functionalities in chaperone regulation [23], although with differing mechanisms [32]. Yet, more recent evidence has shown little similarity in the sequences of the TPR domains of DnaJC7 and Hop, and when combined with the high conservation of the DnaJC7 TPR domain sequence, this may suggest that DnaJC7 has additional or differing functionality from Hop and other TPR domain-containing proteins [21].

## 4. Identification of a Gene–Disease Association between *DNAJC7* and ALS

*DNAJC7* was first identified as a gene associated with ALS in the largest ALS case–control study to date, which utilized whole exome sequencing (WES) data to perform exome-wide variant enrichment and gene burden analyses [3]. The ALS exome sequencing data were gathered as a part of the ALS Genetics (ALSGENS) and Familial ALS (FALS) consortia projects, and the analysis included 3864 ALS patients of European descent. The study replicated rare variant enrichment signals from known ALS genes including *SOD1*, *NEK1*, and *FUS*. Additionally, four carriers of rare, protein truncating variants in *DNAJC7* were observed in the initial ALS cases, whereas zero were observed across the control cohort (*n* = 28,910). Following this observation, an additional 1231 ALS patients were assessed and four other PTV carriers were identified, resulting in a genome-wide significant signal of variant enrichment across the pooled ALS cohort (*n* = 5095). In total, six distinct protein-truncating variants were observed across the eight individuals including *p*.E33X, *p*.Q120X, *p*.R156X, *p*.F163fs, *p*.R216X, and c.1011-2A > G. Additionally, 15 rare missense variants were identified across the ALS cohort in *DNAJC7*, four of which were predicted to be pathogenic in ALS cases, namely, *p*.D8H, *p*.D211N, *p*. R412W, and *p*.E425K. All variants observed were exceedingly rare in the general population. Detailed annotations of all previously reported likely damaging *DNAJC7* variants identified across the various ALS cohorts are reported in Table 2. Since the discovery of *DNAJC7*, multiple cohorts of Asian descent have also been screened for variants of interest in the gene. In total, six additional missense variants have been reported in these cohorts, of which four were predicted to be pathogenic (*p*.R156Q, *p*.K137R, *p*.R238G, and *p*.Y338N), and two were predicted to be benign (*p*.S94T and *p*.N369T) based on variant frequencies and in silico prediction methods [27,28,29]. Additionally, a novel splicing variant (c.754-3T > C) was identified in a sporadic ALS patient of Chinese descent [28], and a novel protein-truncating frameshift variant (*p*.Q134Rfs*6) was identified in a sporadic ALS patient of Taiwanese descent [26]. Figure 1 represents a schematic of the DnaJC7 protein to display the location of all previously reported likely damaging variants.

## 5. Transcriptional Regulation of *DNAJC7*

The heat shock response (HSR), driven by the entire family of heat shock proteins (Hsps), is the cell response to stress and ensures that proteins are properly folded otherwise assisting with re-folding or protein degradation. It is unsurprising that a disease largely driven by misfolded proteins and protein aggregates, such as ALS, has been associated with defects in the HSR [33,34,35]. As described, DnaJC7 is a member of the Hsp protein family and displays many interactions with other proteins involved in the HSR; therefore, gaining a greater understanding of its involvement in the HSR may provide important context regarding its involvement in ALS.

Hsf1 is the major heat shock transcription factor that activates the HSR due to the fact of protein misfolding [36]. Upon accumulation of misfolded proteins in the cytosol and the nucleus, Hsf1 binds to heat shock elements (HSEs) within the promoter region of target genes as a trimer and activates their expression [37,38]. To identify putative Hsf1 binding motifs, we performed promoter region analysis of human *DNAJC7* using the Eukaryotic Promoter Database (EPD) Search Motif Tool and JASPAR CORE 2018 motif library [39,40]. The promoter region was chosen based on the transcription start site common for most *DNAJC7* isoforms (DNAJC7_1). This analysis suggests the presence of a single Hsf1 recognizable HSE, 323 base pairs from the transcription start site (*p* = 0.001). There is also experimental evidence indicating that *DNAJC7* is regulated by Hsf1. First, the ChIP-Seq studies performed as part of the ENCODE project identified a putative Hsf1 binding peak in the *DNAJC7* promoter in a human lymphoblastoid cell line (ENCSR009MBP) and, to a lesser extent, in an MCF-7 human breast cancer cell line (ENCSR062HDL) [41,42]. Additionally, a study by Mendillo et al. using ChIP-Seq found that upon 42 °C heat shock, Hsf1 was abundantly bound in the promoter of *DNAJC7* in primary immortalized mammary epithelial (HME/BPE) and non-malignant breast epithelial (MCF10A) cells but not in tumorigenic BT20, NCIH838, and SKBR3 cell lines [43]. Similarly, *DNAJC7* was not induced by heat shock in HEP-G2 cells, based on Western blotting experiment, mouse embryonic fibroblasts, or PRO-Seq experiment [44,45], and the *DNAJC7* promoter was not bound by Hsf1 in healthy or apoptotic rat cerebral granule neurons [46].

Another hallmark of neurodegenerative disease, and ALS specifically, is oxidative stress caused by the accumulation of reactive oxygen species (ROS) due to the imbalance between the levels of endogenously produced ROS or environmental conditions and the mechanisms that remove them [47,48,49]. Nrf2 is the master transcription factor involved in the oxidative stress response [50]. Upon oxidative stress, Nrf2 dissociates from its negative regulator Keap1 and binds to cis-regulatory sequences known as antioxidant response elements (AREs) [51]. To determine whether *DNAJC7* may be regulated by Nsf1, we again performed promotor analysis using the EPD Search Motif Tool and JASPAR CORE 2018 motif library and identified putative Nrf2 motifs in *DNAJC7* at −508 and −648 base pairs from the transcription start site (*p* = 0.001), although the experimental evidence contradicts the notion that *DNAJC7* is regulated by Nrf2 including a lack of interaction between the protein and *DNAJC7* promotor and a lack of *DNAJC7* induction following oxidative stress [52,53,54].

Finally, we investigated the potential regulation of *DNAJC7* by Nrf1, a transcription factor closely related to Nrf2 that has displayed neuroprotective effects [55,56,57]. While Nrf1 also binds to AREs, it has a slightly differing binding motif to Nrf2 and is known to regulate a distinct subset of genes including those related to the proteosome and cell proliferation [58,59]. Nrf1 possesses several distinct isoforms that regulate different subsets of genes and interact with different co-factors, adding an additional level of complexity to Nrf1 regulation [59,60]. Promoter analysis of *DNAJC7* identified a strong putative binding motif for Nrf1 at –165 base pairs from the transcription start site (*p* = 0.0001). However, similar to Nrf2, the available experimental evidence does not support regulation of *DNAJC7* by Nrf1 including evidence of a lack of interaction with the promotor and lack of induction of *DNAJC7* [59,61,62]. Given the complexity of Nrf1 regulation, it is possible that these studies just did not utilize a unique set of conditions under which Nrf1 regulates *DNAJC7*.

Combined, our analysis of available experimental data suggests that *DNAJC7* may be regulated by Hsf1, possibly in a cell-type-dependent and stress-dependent manner. Previously, inactivation of Hsf1 was shown to result in the accumulation of insoluble, hyperphosphorylated TDP-43 aggregates. In contrast, upregulation of Hsf1 resulted in the increased solubility of SOD1 and improved clearance of insoluble TDP-43 aggregates in a manner mediated by the DnaJC7 interactors, Hsp70 and DnaJB2 [33,63]. While it seems unlikely that *DNAJC7* is regulated by the master transcription factor of oxidative stress response, Nrf2, it is possible that under some conditions, it may be regulated by Nrf1, and this mode of regulation may warrant further investigation.

## 6. Expression of *DNAJC7* in the Human Central Nervous System

Of the 49 J proteins encoded within the human genome, the majority are expressed in the brain at varying levels, although *DNAJC7* is considered highly expressed [64]. Using the expression of the genes encoding Hsp70 and Hsp90 proteins as a baseline, we compared the expression of multiple J protein-encoding genes—including those associated with neurological conditions according to ClinVar (Table 1), among others—which demonstrated a wide range of expression levels throughout the human central nervous system (CNS) (Figure 3). All expression data were obtained from the Genotype-Tissue Expression (GTEx) database (https://gtexportal.org/home/ accessed on 12 March 2022) [65].

The Hsp70 protein family are encoded by *HSPA* genes and are the most abundant chaperones, as they carry out a variety of functions including protein folding, translocation across organelle membranes, and prevention of aggregate formation. Here, we analyzed one constitutive and one stress-induced *HSPA* as a reference for molecular chaperone expression. The constitutive *HSPA2* is expressed at moderate to high levels throughout the CNS with notably high expression in the spinal cord, whereas the stress-induced *HSPA12A* featured moderate to low expression in the human brain and spinal cord. Similarly, Hsp90s are ubiquitously expressed at very high levels. Overall, throughout the CNS, *DNAJC7* displays higher expression levels than the Hsp70s but lower overall expression than the Hsp90s. The J protein’s expression levels were most like that of *HSPA2* and most distinct to those of *HSP90AB1* and *HSP90AA1*, although the differences were marginal.

Of the J protein-encoding genes, the mostly highly expressed throughout the CNS included *DNAJA1* and *DNAJB2*, although *DNAJC5*, *DNAJC6*, *DNAJB1*, and *DNAJC7* were all also considered to be highly expressed across the brain regions. Contrarily, many other DNAJCs were among those most lowly expressed throughout the CNS such as *DNAJC28*, *SACS*, and *DNAJC16*. Yet, many of these DNAJs have been previously associated with neurological phenotypes within the ClinVar database (Table 1). Overall, this expression analysis suggests a prominent role of DnaJC7 in the human CNS and indicates that loss of DnaJC7 function in the CNS could be particularly detrimental and cause ALS-associated neurodegeneration.

## 7. Interactions of DnaJC7 with HSP70s, HSP90s, and Other Proteins

DnaJC7 has demonstrated interactions with Hsp70s and Hsp90s as well as with important co-chaperones of these molecular chaperone systems [66]. Here, we analyzed known physical interactions between a selection of J proteins, Hsp70s, and Hsp90s and describe notable interactions with DnaJC7 that are potentially relevant to the encoding gene’s association with ALS. Figure 4 displays an interaction map created using the GeneMANIA prediction server to visualize the physical interactions between these proteins (https://genemania.org/ accessed on 12 March 2022) [67].

Expectedly, DnaJC7 displays physical interactions with Hsp70s, including HSPA2, HSPA4, and HSPA8, and Hsp90s such as HSP90AA1 and HSP90AB1 [23,32]. Interactions with the Hsp70s are mediated by DnaJC7’s conserved J-domain, consistent with the J-domain functionality of other J proteins [23]. Although this interaction is independent of any other protein features, evidence does suggest that the TPR domains may have stabilizing effects [32]. The interaction stimulates the ATPase function of Hsp70s, allowing the protein class to perform stable polypeptide binding and protein folding [23]. Notably, HSPA8 was shown to be reduced in the primary motor neurons and neuromuscular junctions in TDP-43-mediated mouse models of ALS as well as in C9orf72-mediated fly models and human-induced pluripotent stem cells [68]. DnaJC7 binds Hsp90s through its TPR domains and has a role in the disruption of interactions between Hsp90s and its substrates, allowing for proteins to re-enter the protein folding pathway if not properly folded after a single pass through the Hsp70–Hsp90 folding system [23,32]. This suggests that while DnaJC7 may increase the efficiency of protein folding, if overexpressed, it may prevent the completion of protein folding by Hsp90s [23]. Of particular interest, HSP90 has previously demonstrated an ability to bind to TDP-43 and contribute to its clearance; however, the clearance is inhibited when aberrant tau accumulates in the cytosol [69,70,71]. Recently, DnaJC7 was found to bind and stabilize natively folded conformations of tau [21]. While it remains unknown whether this function is compromised by ALS-associated *DNAJC7* variants, we hypothesize that this may be a potential functional mechanism of the variants, such that mutant DnaJC7 results in tau accumulation and downstream inhibition of TDP-43 clearance.

DnaJC7 also demonstrates physical interactions with non-Hsp70/Hsp90 chaperones, including with the other J proteins DnaJB1 and DnaJB2, although the exact mechanism and function of the interactions remain unclear. Of note, overexpression of both DnaJBs have shown protective effects in models of ALS. Specifically, overexpression of DnaJB1, and its yeast homolog Sis1, reduced TDP-43-mediated toxicity including reduced effects on cell growth, cell shape, and ubiquitin–proteasome system inhibition [72]. Similarly, overexpression of DnaJB2 improved outcomes of mutant SOD1 in in vivo ALS models including improvements in muscle performance and motor neuron survival [20]. DnaJC7 also interacts with a selection of other proteins including RAD1, RAD9A, and STUB1. Together, RAD1 and RAD9A form a complex responsible for cell cycle checkpoint signaling [73] and were both found to bind to DnaJC7 via its TPR domain; however, new studies suggest that the J-domain has a role in the regulation of the interaction with RAD9 including influencing complex formation and localization [74]. Finally, similarly to DnaJC7, STUB1 is an Hsp90 co-chaperone protein involved in protein quality control, and it contains multiple TPR domains. Although there is not a thorough understanding of this interaction, in complex with Hsp70, STUB1 has displayed neuroprotective effects by modulating the degradation of mutant SOD1 [75]. The interaction and similarities between DnaJC7 and STUB1 suggest further investigation regarding DnaJC7’s involvement in the SOD1 pathway is warranted in the context of fALS caused by mutations in SOD1.

The rather striking number of associations between DnaJC7 interactors and ALS provides promise that interrogating these interactions further may allow for greater understanding of its general role in cellular protein quality control and the pathological mechanism underlying the association of *DNAJC7* with ALS.

## 8. Possible Role of Pathogenic *DNAJC7* Variants in ALS

It is plausible that some ALS-associated mutations in *DNAJC7* disrupt or even completely abolish the chaperone function of DnaJC7, supporting a loss-of-function mechanism. In the analyses presented by Farhan et al., immunoblot assays of DnaJC7 from a fibroblast sample of an ALS patient carrying a protein-truncating variant (*p*.R156X) identified that the variant resulted in significantly reduced protein levels [3]. Yet, further mechanistic studies are warranted to provide conclusive evidence.

It remains unclear why the variants in *DNAJC7* result in ALS pathogenesis, i.e., mostly affecting motor neurons and no other neurons or non-neuronal tissues. Ablation of DnaJC7 expression in mice produces viable animals that seem to develop normally, indicating non-essential functions of DnaJC7 at least under normal conditions [76]. Further, recent evidence suggests DnaJC7 binds and stabilizes natively folded tau, the dysmetabolism of which is observed in frontotemporal spectrum disorder of ALS (ALS-FTSD) [77]. We propose that DnaJC7 becomes essential under specific stress conditions, such as oxidative stress, protein misfolding stress, and/or the diminished function of protein quality control in aged cells, specifically in motor neurons. Additionally, misfolded proteins commonly associated with ALS, such as TDP-43, FUS, and SOD1, might require particularly diligent chaperoning by Hsp70 and Hsp90, which may be regulated by DnaJC7. Finally, there is previous evidence that the homolog of *DNAJC7* in Drosophila may suppress polyglutamine-induced toxicity [78,79]. Although these studies were not directly in relation to *ATXN2*, the gene known to carry CAG-repeat risk factors for ALS, further investigation into this mechanism and the *ATXN2* polyglutamine status of *DNAJC7* variant carriers may prove beneficial.

## 9. Conclusions

In this review article, we summarized the evolutionary conservation, expression profile, and proposed function in cellular protein quality control of the J protein, DnaJC7, and discussed how potentially pathogenic variants in *DNAJC7* can contribute to neurodegeneration in ALS, possibly via a loss-of-function mechanism. Based on the collected evidence, it seems that *DNAJC7* regulation may be controlled in response to cell stress via the HSR under conditions that are yet to be defined. However, when mutated, we hypothesize that there may be dysregulation of downstream pathways. These may include the dysregulation of tau, potentially resulting in the accumulation of TDP-43 through the inhibition of proper HSP90AB1 function, destabilization of interactions with other J proteins involved in neuroprotection, or the inability to accurately chaperone the variety of Hsp70s or Hsp90s involved in ALS-associated protein aggregates; although, other potential mechanisms cannot be ruled out. Future studies are required to determine the functions of DnaJC7 under normal conditions and under cellular stress conditions, its specific clients, and how DnaJC7 interacts with and processes misfolded proteins, specifically those that characterize ALS.

## Figures and Tables

**Figure 1 ijms-23-04076-f001:**
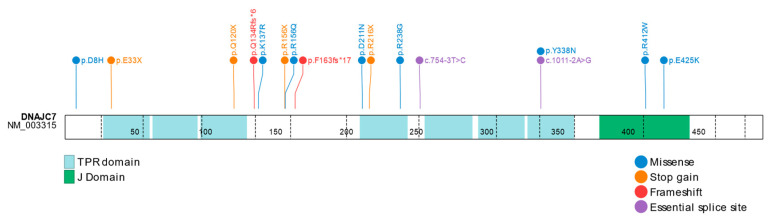
Protein schematic of the previously reported likely damaging variants in *DNAJC7* in various ALS cohorts. Variants were previously reported to be associated with ALS by Farhan et al.; Jih et al.; Wang et al.; He et al.; Sun et al. [3,26,27,28,29]. Exon boundaries are indicated by black vertical, dashed lines.

**Figure 2 ijms-23-04076-f002:**
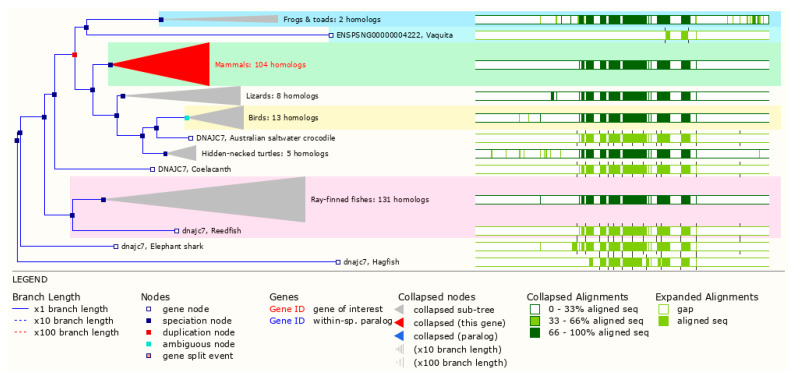
Phylogenetic tree of human *DNAJC7* including protein sequence alignment. The single gene split event at *P. reticulata* is not shown. The phylogenetic tree was built using the Gene Tree application within the current release of Ensembl [30].

**Figure 3 ijms-23-04076-f003:**
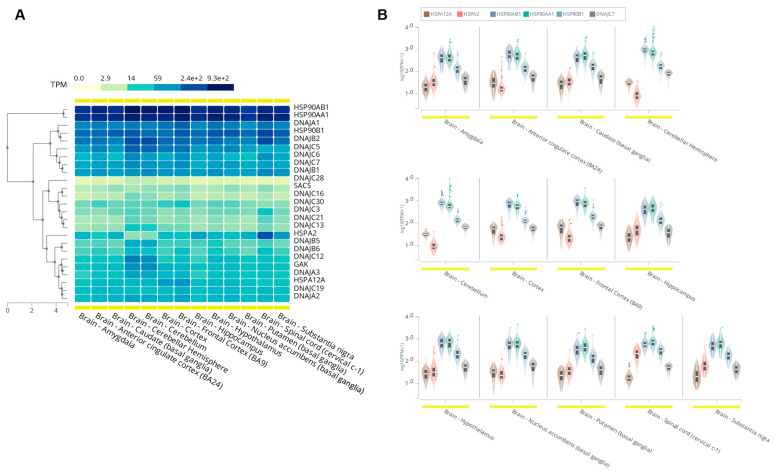
Expression of various J protein-, Hsp70-, and Hsp90-encoding genes throughout the human central nervous system (CNS). Gene expression data were obtained from Genotype-Tissue Expression (GTEx) database (https://gtexportal.org/home/ accessed on 12 March 2022) [65]. (**A**) Expression levels of various J protein-, Hsp70-, and Hsp90-encoding genes throughout the human CNS. The tree along the left *y*-axis demonstrates the relative difference between the various genes’ overall CNS expression profiles; (**B**) violin plots comparing the expression levels of *DNAJC7* to a sample of Hsp70- and Hsp90-encoding genes.

**Figure 4 ijms-23-04076-f004:**
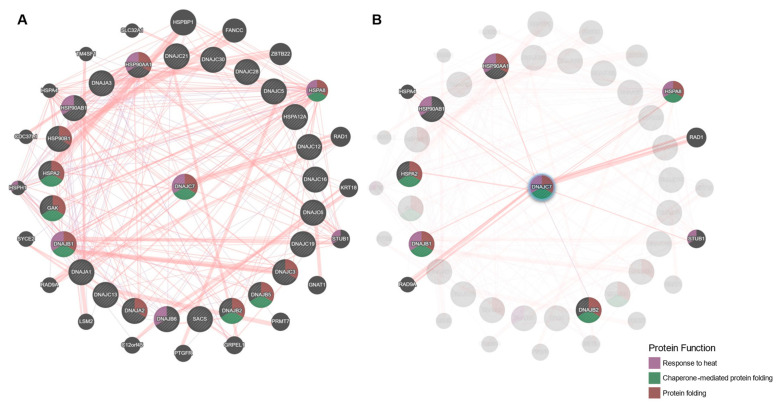
Map of known physical interactions between a selection of J proteins, Hsp70s, Hsp90s, and other relevant co-factors. The interaction map was created using the GeneMANIA prediction server (https://genemania.org/ accessed on 12 March 2022) [67] by inputting a selection of 20 J proteins, two Hsp70s, and three Hsp90s. Using this selection, the prediction server was then able to select other relevant interactors: (**A**) all physical interactions between the J proteins, Hsp70s, Hsp90s, and other relevant cofactors; (**B**) proteins known to physically interact with DnaJC7.

**Table 1 ijms-23-04076-t001:** Number of variants in J proteins, otherwise referred to as Hsp40s, previously associated with neurological phenotypes and/or phenotypes presenting with neurological features.

Gene	Disease	ClinVar Pathogenic		ClinVar Likely Pathogenic	
		Missense	LoF	Missense	LoF
*DNAJA3*	Developmental delay and polyneuropathy	NA	NA	NA	NA
*DNAJB2*	Charcot–Marie–Tooth disease; distal spinal muscular atrophy	NA	7	2	4
*DNAJB5*	Peripheral neuropathy; skeletal myopathy; peripheral neuropathy	NA	NA	1	NA
*DNAJB6*	Limb–girdle muscular dystrophy, type 1E; frontotemporal dementia	8	1	4	NA
*DNAJC3*	Combined cerebellar and peripheral ataxia with hearing loss and diabetes mellitus	NA	3	NA	1
*DNAJC5*	Neuronal ceroid lipofuscinosis	1	2	NA	NA
*DNAJC6*	Juvenile-onset Parkinson’s disease 19a	2	9	NA	NA
*DNAJC7*	Amyotrophic lateral sclerosis	NA	NA	NA	NA
*DNAJC12*	Mild hyperphenylalaninemia, non-bh4-deficient; early-onset parkinsonism	2	10	NA	2
*DNAJC13*	Late-onset Parkinson’s disease; essential tremor	1	NA	NA	NA
*DNAJC16*	Hereditary spastic paraplegia	NA	NA	NA	NA
*DNAJC19*	3-Methylglutaconic aciduria type V	NA	4	1	2
*DNAJC21*	Bone marrow failure syndrome 3; tongue abnormality, acute myeloid leukemia, cognitive impairment, pancytopenia, pectus excavatum, short stature, and webbed neck	2	8	NA	NA
*DNAJC28*	Delayed speech and language, generalized hypotonia, intellectual disability, seizures, and optic atrophy	NA	NA	NA	NA
*DNAJC30*	Leber hereditary optic neuropathy	3	NA	1	NA
*GAK*	Parkinson’s disease	NA	NA	NA	NA
*SACS*	Spastic ataxia of Charlevoix–Saguenay; Spastic paraplegia	18	155	17	188

Previous disease associations were based on reports of variant pathogenicity in the ClinVar database. NA: no variants were reported with the pathogenicity classification in ClinVar; LoF: putative loss-of-function variants, including essential splice site, frameshift, stop gain, and stop loss variants.

**Table 2 ijms-23-04076-t002:** Likely damaging variants previously identified in *DNAJC7* in multiple ALS cohorts.

cDNA Change	Protein Change	Variant Type	ALS Cases (*N*)	GnomAD (Non-Neuro v2.1.1) MAF	In Silico Prediction (CADD)	Reference
c.22G > C	*p*.D8H	Missense	1 (5095)	0.0000198	25	Farhan et al., 2019 [3]
c.97G > T	*p*.E33X	Stop gain	1 (5095)	0	39	Farhan et al., 2019 [3]
c.358C > T	*p*.Q120X	Stop gain	1 (5095)	0	37	Farhan et al., 2019 [3]
c.401_402delAA	*p*.Q134Rfs*6	Truncating frameshift	1 (325)	0	31	Jih et al., 2020 [26]
c.410A > G	*p*.K137R	Missense	1 (326)	0	23	Sun et al., 2021 [29]
c.466C > T	*p*.R156X	Stop gain	2 (5095)	0	41	Farhan et al., 2019 [3]
c.467G > A	*p*.R156Q	Missense	1 (701)	0.0000146	23	He et al., 2021 [28]
c.488delT	*p*.F163fs*17	Frameshift	1 (5095)	0	NA	Farhan et al., 2019 [3]
c.631G > A	*p*.D211N	Missense	1 (5095)	0	26	Farhan et al., 2019 [3]
c.646C > T	*p*.R216X	Stop gain	2 (5095)	0	40	Farhan et al., 2019 [3]
c. 712A > G	*p*.R238G	Missense	1 (578)	0	24	Wang et al., 2020 [27]
c.754-3T > C	NA	Essential splice site	1 (701)	0.0000244	15	He et al., 2021 [28]
c.1011-2A > G	NA	Essential splice site	1 (5095)	0	26	Farhan et al., 2019 [3]
c.1012T > A	*p*.Y338N	Missense	1 (701)	0	28	He et al., 2021 [28]
c.1234C > T	*p*.R412W	Missense	1 (5095)	0.0000040	34	Farhan et al., 2019 [3]
c.1273G > A	*p*.E425K	Missense	2 (5095)	0	35	Farhan et al., 2019 [3]

ALS, amyotrophic lateral sclerosis; CADD, combined annotation-dependent depletion; MAF, minor allele frequency; *N*, total cohort size; NA, not applicable.

## Data Availability

Not applicable.

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
