# Peer review of "DnaJC7 in Amyotrophic Lateral Sclerosis"

_ijms, 2022, doi:10.3390/ijms23084076_

Round 1
Reviewer 1 Report
Like other neurological disorders, ALS is also characterized by aggregation and accumulation of misfolded protein in cells. Chaperones have central roles in maintaining protein homeostasis by preventing protein aggregation and directing the terminally misfolded protein to degradation. The close association between various chaperones and co-chaperones is essential to prevent protein aggregation. In this line, the review by Dilliott et al on DNAJC7 is very exciting and informative. The study and data are well designed and presented. I have a couple of minor suggestions that the author can consider including
- lines 69-83; is basically the Chaperon cycle. A figure can be very helpful to readers new to the field.
- There are redundancies in information at several places such as lines 117-119 and 135-138. This is just an example. Please check other places as well.
Author Response
Wew would like to thank the reviewer for their positive assessment of our review article.
- lines 69-83; is basically the Chaperon cycle. A figure can be very helpful to readers new to the field.I am not sure if we should add such a figure…this has been reviewed excessively in the past…
Our esponse: We appreciate the reviewer pointing out that this is indeed the basics of the chaperone cycle; however, the cycle has been extensively reviewed in the literature previously and because it is not the focus of our review, we do not feel a figure will add to the overall flow of our manuscript. Instead, we have added a line within the paragraph indicated stating that the J protein/Hsp70 ATPase cycle is “an example of a Chaperone cycle” so that the reader may pursue further reading about the topic if they so wish.
2. There are redundancies in information at several places such as lines 117-119 and 135-138. This is just an example. Please check other places as well.
Our response: We thank the reviewer for the careful identification of this redundancy. We have removed the reiteration of the function of the TPR domains specified and have completed a thorough review of the manuscript for any additional redundancy.
Reviewer 2 Report
DnaJs o Hsp40 chaperones are a large group of molecular chaperones considered in the past to have a secondary role, mostly as cochaperones of Hsp70. However, this view has dramatically changed, and currently they are considered to have important and primary roles in many different processes, this due in part to the many different types of DnaJs present in higher organisms. There are three classes of DnaJs, and a large part of them correspond to the least known class III, which only have in common the presence of J domain (responsible for the interaction with Hsp70). This is the case of DnaJC7, which hosts a J domain and three tetratricopeptide repeats (TPR) an arrangement very similar to that of cochaperone Hop, which have been shown to be used to link DnaJC7 with Hsp70 (J domain) and Hsp90 (through the TPR domain).
This very good short review by Dilliott et al. summarizes the known functions of DnaJC7 in protein quality control and and even more interesting, includes a discussion on how potentially pathogenic variants in the DNAJC7 gene (this has been studied extensively) can contribute to neurodegeneration in amyotrophic lateral sclerosis (ALS).
Comments:
- I strongly urge you to merge Fig. 1 and Fig. 3 (no much difference between the two)
- Given the similar arrangement of the three DNAJC7 TPR domains to those of Hop, and the fact that Hop binds to Hsp70 through either TPR1 and TPR2B, it might look as if both cochaperones could do similar functions, and the papers by Brychzy et al. (2003) and Moffat et al. (2008) seem to support this view. However, Huo et al. (2021) find the sequence of the three TPR domains quite different (Discussion section), which probably argues about Hop being a Hsp70/Hsp90 linker. Perhaps a comment about this should be included in this manuscript
- It would be interesting to describe (if it is known) what are the effects of the pathogenic mutations; do they give rise to properly folded molecules?
Some other comments:
- Line 172: Figure 2 should be Fig. 3 (although see my comment above)
- Line 256: (Figure 3) should be (Figure 4)
- Line 291: Figure 4 should be Figure 5
- Lines 318-320: The statement “DnaJC7 also demonstrates physical interactions with non-Hsp70/Hsp90 chaperones, including with the other J proteins DnaJB1 and DnaJB2, although the exact mechanism and function of the interactions remain unclear” should be supported with a reference.
- Lines 346 and 347: A and B should be in bold.
- Lines 356-358: The statement “Ablation of DnaJC7 expression in mice produces viable animals that seem to develop normally, indicating non-essential functions of DnaJC7 at least under normal conditions.” should be supported with a reference
Author Response
We would like to thanks the reviwiewer for their thoughtul comments and suggestions.
- Given the similar arrangement of the three DNAJC7 TPR domains to those of Hop, and the fact that Hop binds to Hsp70 through either TPR1 and TPR2B, it might look as if both cochaperones could do similar functions, and the papers by Brychzy et al. (2003) and Moffat et al. (2008) seem to support this view. However, Huo et al. (2021) find the sequence of the three TPR domains quite different (Discussion section), which probably argues about Hop being a Hsp70/Hsp90 linker. Perhaps a comment about this should be included in this manuscript
Our response: We agree that this information could strengthen the discussion of DnaJC7 conservation and resulting functionality and have added a paragraph on lines 142-152 to summarize the available information in the literature. Thanks for the great suggestion.
Also, as suggested, we removed Figure 1 as it is indeed highly similar to Figure 3
References were added as requested by the reviewer and all other errors were corrected.